# QuadMamba: Learning Quadtree-based Selective Scan for Visual State Space Model

**Fei Xie**[1]    **Weijia Zhang**[1]    **Zhongdao Wang**[2]    **Chao Ma**[1]*

[1] MoE Key Lab of Artificial Intelligence, AI Institute, Shanghai Jiao Tong University
[2] Huawei Noah's Ark Lab

{jaffe031, weijia.zhang, chaoma}@sjtu.edu.cn
wangzhongdao@huawei.com

## Abstract

Recent advancements in State Space Models, notably Mamba, have demonstrated superior performance over the dominant Transformer models, particularly in reducing the computational complexity from quadratic to linear. Yet, difficulties in adapting Mamba from language to vision tasks arise due to the distinct characteristics of visual data, such as the spatial locality and adjacency within images and large variations in information granularity across visual tokens. Existing vision Mamba approaches either flatten tokens into sequences in a raster scan fashion, which breaks the local adjacency of images, or manually partition tokens into windows, which limits their long-range modeling and generalization capabilities. To address these limitations, we present a new vision Mamba model, coined QuadMamba, that effectively captures local dependencies of varying granularities via quadtree-based image partition and scan. Concretely, our lightweight quadtree-based scan module learns to preserve the 2D locality of spatial regions within learned window quadrants. The module estimates the locality score of each token from their features, before adaptively partitioning tokens into window quadrants. An omnidirectional window shifting scheme is also introduced to capture more intact and informative features across different local regions. To make the discretized quadtree partition end-to-end trainable, we further devise a sequence masking strategy based on Gumbel-Softmax and its straight-through gradient estimator. Extensive experiments demonstrate that QuadMamba achieves state-of-the-art performance in various vision tasks, including image classification, object detection, instance segmentation, and semantic segmentation. The code is in https://github.com/VISION-SJTU/QuadMamba.

## 1 Introduction

The architecture of Structured State Space Models (SSMs) has gained significant popularity in recent times. SSMs offer a versatile approach to sequence modeling that balances computational efficiency with model flexibility. Inspired by the success of Mamba [13] in language tasks, there has been a rise in using SSMs for various vision tasks. These applications range from designing generic backbone models [80, 41, 26, 66, 49] to advancing fields such as image segmentation [51, 39, 65, 46] and synthesis [17]. These advancements highlight the adaptability and potential of Mamba in the visual domain.

Despite their appealing linear complexity in long sequence modeling, applying SSMs directly to vision tasks results in only marginal improvements over prevalent CNNs and vision Transformer

---

*Corresponding author

38th Conference on Neural Information Processing Systems (NeurIPS 2024).

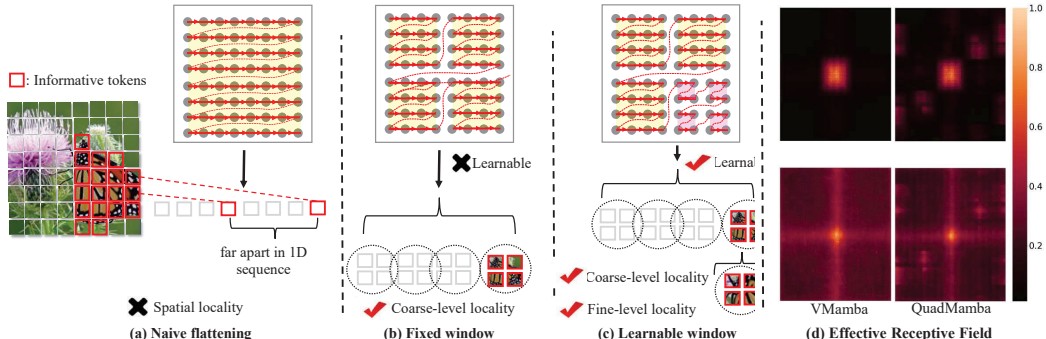

Figure 1: Illustration of scan strategies for transforming 2D visual data into 1D sequences. (a) naive raster scan [80, 41, 66] ignores the 2D locality; (b) fixed window scan [26] lacks the flexibility to handle visual signals of varying granularities; (c) our learnable window partition and scan strategy adaptively preserves the 2D locality with a focus on the more informative window quadrant; (d) the effective receptive field of our QuadMamba demonstrates more locality than the plain Vision Mamba.

models. In this paper, we seek to expand the applicability of the Mamba model for computer vision. We observe that the differences between the language and visual domains can pose significant obstacles in adapting Mamba to the latter. The challenges come from two natural characteristics of image data: 1) Visual data has rigorous 2D spatial dependencies, which means flattening image patches into a sequence may destroy the high-level understanding. 2) Natural visual signals have heavy spatial redundancy—e.g., an irrelevant patch does not influence the representation of objects. To address these two issues, we develop a vision-specific scanning method to construct 1D token sequences for Vision Mamba. Typically, vision Mamba models need to transform 2D images into 1D sequences for processing. As illustrated in Fig. 1(a), the straightforward method, e.g., Vim [80], that flattens spatial data into 1D tokens directly disrupts the natural local 2D dependencies. LocalMamba improves local representation by partitioning the image into multiple windows, as shown in Fig. 1(b). Each window is scanned individually before conducting a traversal across windows, ensuring that tokens within the same 2D semantic region are processed closely together. However, the handcrafted window partition lacks the flexibility to handle various object scales and is unable to ignore the less informative regions.

In this work, we introduce a novel Mamba architecture that learns to improve local representation by focusing on more informative regions for locality-aware sequence modeling. As shown in Fig. 1(c), the gist of QuadMamba lies in the learnable window partition that adaptively learns to model local dependencies in a coarse-to-fine manner. We propose to employ a lightweight prediction module to multiple layers in the vision Mamba model, which evaluates the local adjacency of each spatial token. The quadrant with the highest score is further partitioned into sub-quadrants in a recursive fashion for fine-grained scan, while others, likely comprising less informative tokens, are kept in a coarse granularity. This process results in window quadrants of varying granularities partitioned from the 2D image feature.

It is noteworthy that direct sampling from the 2D windowed image feature based on the index is non-differentiable, which renders the learning of window selection intractable. To handle this, we adopt Gumbel-Softmax to generate a sequence mask from the partition score maps. We then employ fully differentiable operators, i.e., Hadamard product and element-wise summation, to construct the 1D token sequences from the sequence mask and local windows. These lead to an end-to-end trainable pipeline with negligible computational overhead. For the informative tokens that cross two adjacent quarter windows, we apply an omnidirectional shifting scheme in successive blocks. Shifting 2D image features in two directions allows the quarter-window partition to be flexible in modeling objects appearing in arbitrary locations.

Extensive experiments on ImageNet-1k and COCO2017 demonstrate that QuadMamba excels at image classification, object detection, and segmentation tasks, with considerable advantages over existing CNN, Tranformer, and Mamba models. For instance, QuadMamba achieves a Top-1 accuracy of 78.2% on ImageNet-1k with a similar model size as PVT-Tiny (75.1%) and LocalViM (76.2%).

## 2 Related Work

### 2.1 Generic Vision Backbones

Convolutional Neural Networks (CNNs) [10, 30, 31] and Vision Transformer (ViT) [7] are two categories of dominant backbone networks in computer vision. They have proven successful as a generic vision backbone across a broad range of computer vision tasks, including but not limited to image classification [29, 53, 55, 20, 23, 21, 74, 4, 12], segmentation [44, 19], object detection [36, 79], video understanding [28, 76], and generation [11]. In contrast to the constrained receptive field in CNNs, vision Transformers [7, 42, 61], borrowed from the language tasks [59], are superior in global context modelling. Later, numerous vision-specific modifications are proposed to adapt the Transformer better to the visual domain, such as the introduction of hierarchical features [42, 61], optimized training [58], and integration of CNN elements [5, 54]. Thus, vision Transformers demonstrate leading performance on various vision applications [63, 1, 8, 34, 48]. This, however, comes at a cost of attention operations' quadratic time and memory complexity. which hinders their scalability despite remedies proposed [42, 61, 72, 57].

More recently, State Space Models (SSMs) emerged as a powerful paradigm for modeling sequential data in language tasks. Advanced SSM models [13] have reported even superior performance compared to state-of-the-art ViT architectures while having a linear complexity. Their initial success on vision tasks [80, 41] and, more importantly, remarkable computational efficiency hint at the potential of SSM as a promising general-purpose backbone alternative to CNNs and Transformers.

### 2.2 State Space Models

State Space Models (SSMs) [16, 15, 18, 35] are a family of fully recurrent architectures for sequence modeling. Recent advancements [15, 14, 47, 13] have gained SSMs Transformer-level performance, yet with its complexity scaling linearly. As a major milestone, Mamba [13] revamped the conventional SSM with input-dependent parameterization and scalable, hardware-optmized computation, performing on par with or better than advanced Transformer models on different tasks involving sequential 1D data.

Following the success of Mamba, ViM [80] and VMamba [41] reframe Mamba's 1D scan into bi-directional and four-directional 2D cross-scan for processing images. Subsequently, SSMs have been quickly applied to vision tasks (semantic segmentation [51, 65, 46], object detection [26, 3], image restoration [17, 52], image generation [9], etc.) and to data of other modalities (e.g., videos [67, 32], point clouds [40, 73], graphs [2], and cross-modality learning [60, 6]).

A fundamental consideration in adapting Mamba to non-1D data concerns the design of a path that scans through and maps all image patches into a SSM-friendly 1D sequence. Along this direction, preliminary efforts include the bi-directional zigzag-style scan in ViM [80], the 4-direction cross-scan in VMamba [41], and the serpentine scan in PlainMamba [66] and ZigMa [22], all conducted in the spatial domain spanned by the height and width axes. Other works [52, 75, 33] extend the scanning to an additional channel [52, 33] or temporal [75, 33] dimension. Yet, by naively traversing the patches, these scan strategies overlooked the importance of spatial locality preservation. This inherent weakness has been partially mitigated by LocalMamba [26], which partitions patches into windows and performs traversal within each window.

However, due to a monolithic locality granularity throughout the entire image domain, as controlled by the arbitrary window size, it is hard to decide on an optimal granularity. LocalMamba opts for DARTS [38] to differentially search for the optimal window size alongside the optimal scan direction for each layer, which adds to the complexity of the method. On the other hand, all existing methods involve hard-coded scan strategies, which can be suboptimal. Unlike all these methods, this paper introduces a learnable quadtree structure to scan image patches with varying locality granularity.

## 3 Preliminaries

**State Space Models (S4)**. SSMs [16, 15, 18, 35] are in essence linear time-invariant systems that map a one-dimensional input sequence $x(t) \in \mathbb{R}^L$ to output response sequence $y(t) \in \mathbb{R}^L$ recurrently through hidden state $h(t) \in \mathbb{R}^N$ (with sequence length $L$ and state size $N$). Mathematically, such

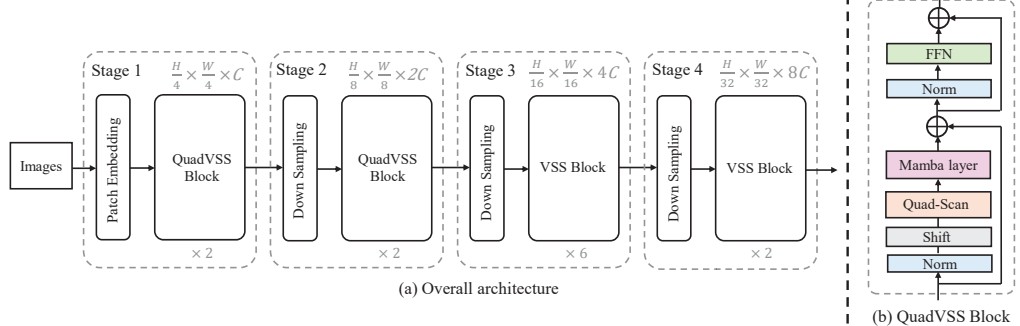

Figure 2: The pipeline of the proposed QuadMamba (a) and its building block: QuadVSS block (b). Similar to the hierarchical vision Transformer, QuadMamba builds stages with multiple blocks, making it flexible to serve as the backbone for vision tasks.

systems can be formulated as the following Ordinary Differential Equations (ODEs):

$$h'(t) = \boldsymbol{A}h(t) + \boldsymbol{B}x(t)$$
$$y(t) = \boldsymbol{C}h(t) \tag{1}$$

where matrix $\boldsymbol{A} \in \mathbb{R}^{N \times N}$ contains the evolution parameters; $\boldsymbol{B} \in \mathbb{R}^{N \times 1}$ and $\boldsymbol{C} \in \mathbb{R}^{1 \times N}$ are projection matrices.

The continuous-time ODEs in Eqn. 1 are, however, difficult to solve using deep learning. In practice, Eqn. 1 is usually transformed into a discretized form [16] via the zero-order hold (ZOH) rule, where the value of $x$ is held constant over a sample interval $\Delta$ [18]. The discretized ODEs are given by:

$$h(t) = \overline{\boldsymbol{A}}h(t-1) + \overline{\boldsymbol{B}}x(t),$$
$$y(t) = \boldsymbol{C}h(t). \tag{2}$$

where $\overline{\boldsymbol{A}}$ and $\overline{\boldsymbol{B}}$ are the discrete version of $\boldsymbol{A}$ and $\boldsymbol{B}$: $\overline{\boldsymbol{A}} = \exp(\Delta\boldsymbol{A})$, and $\overline{\boldsymbol{B}} = (\Delta\boldsymbol{A})^{-1}(\overline{\boldsymbol{A}}-\boldsymbol{I})\cdot\Delta\boldsymbol{B}$. For efficient implementation, the iterative computations in Eqn. 2 can be performed in parallel with a global convolution operation:

$$\boldsymbol{y} = \boldsymbol{x} \circledast \overline{\boldsymbol{K}}$$
$$\text{with} \quad \overline{\boldsymbol{K}} = (\boldsymbol{C}\overline{\boldsymbol{B}}, \boldsymbol{C}\overline{\boldsymbol{A}}\overline{\boldsymbol{B}}, ..., \boldsymbol{C}\overline{\boldsymbol{A}}^{L-1}\overline{\boldsymbol{B}}), \tag{3}$$

where $\circledast$ denotes the convolution operator and $\overline{\boldsymbol{K}} \in \mathbb{R}^L$ the SSM kernel.

**Selective State Space Models (S6)**. Traditional SSMs have input-agnostic parameters. To improve this, Selective State Space Models [13], also referred to as "S6" or "Mamba", are proposed with input-dependent parameters, such that $\overline{\boldsymbol{A}}$, $\overline{\boldsymbol{B}}$, and $\Delta$ become learnable. To make up for parallelism difficulties, hardware-aware optimization is also performed. In this work, we particularly investigate the effective adaptation of the Mamba architecture for vision tasks. Early works such as ViM [80] and VMamba [41] have explored intuitive adaptations by transforming 2D images to 1D sequences in a raster scan fashion. We argue that simple raster scan is not an optimal design as it breaks the local adjacency of images. In our work, a novel learnable scanning scheme based on quadtree is proposed.

## 4 Method

### 4.1 General Architecture

QuadMamba shares a similar multi-scale backbone design to many CNNs [20, 64, 23] and vision Transformers [61, 71, 42, 25]. As shown in Fig. 2, an image $I \in \mathbb{R}^{H_{im} \times W_{im} \times 3}$ is first partitioned into patches of size $4 \times 4$, resulting in $N = H \times W = \lfloor\frac{H_{im}}{4}\rfloor \times \lfloor\frac{W_{im}}{4}\rfloor$ visual tokens. A linear layer maps these visual tokens to hidden embeddings with dimension $d$, which are subsequently fed into our proposed Quadtree-based Visual State Space (QuadVSS) blocks. Unlike the Mamba structure used in language modeling [13], the QuadVSS blocks follow the popular structure of the

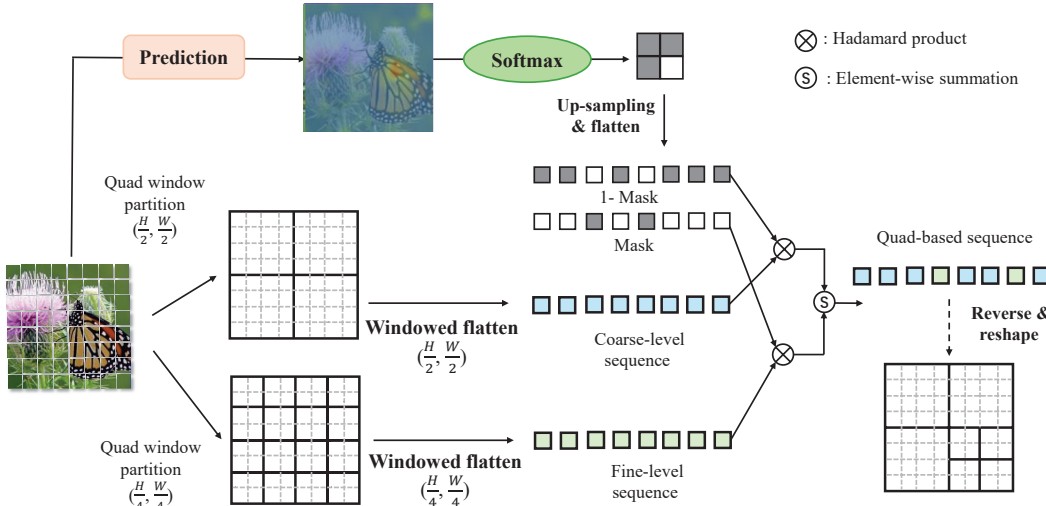

Figure 3: Quadtree-based selective scan with prediction modules. Image tokens are partitioned into bi-level window quadrants from coarse to fine. A fully differentiable partition mask is then applied to generate the 1D sequence with negligible computational overhead.

Transformer block [7, 68], as illustrated in Fig 2(b). QuadMamba consists of a cascade of QuadVSS blocks organized in four stages, with stage $i$ ($i \in \{1, 2, 3, 4\}$) having $S_i$ QuadVSS blocks. In each stage, a downsampling layer halves the spatial size of feature maps while doubling their channel dimension. Thanks to the linear complexity of Mamba, we are free to stack more QuadVSS blocks within the first two stages, which enables their local feature preserving and modeling capabilities to be fully exploited with minimal computational overheads introduced.

## 4.2   Quadtree-based Visual State Space Block

As shown in Fig 2, our QuadVSS block adopts the meta-architecture [68] in vision Transformer, formulated by a token operator, a feedforward network (FFN), and two residual connections. The token operator consists of a shift module, a partition map predictor, a quadtree-based scanner, and a Mamba Layer. Inside the token operator, a lightweight prediction module first predicts a partition map over feature tokens. The quadtree-based strategy then partitions the 2D image space by recursively subdividing it into four quadrants or windows. According to the scores of the partition map at the coarse level, fine-level sub-windows within less informative coarse-level windows are skipped. Thus, a multi-scale, multi-granularity 1D token sequence is constructed, capturing more locality in more informative regions while retaining global context modeling elsewhere. The key components of the QuadVSS block are detailed as follows:

**Partition map prediction.** The image feature $\boldsymbol{x} \in \mathbb{R}^{H \times W \times C}$, containing a total of $N = HW$ embedding tokens, is first projected into score embeddings $\boldsymbol{x}_{\mathrm{s}}$:

$$\boldsymbol{x}_{\mathrm{s}} = \phi_s(\boldsymbol{x}), \quad \boldsymbol{x}_{\mathrm{s}} \in \mathbb{R}^{N \times C}, \tag{4}$$

where $\phi_s$ is a lightweight projector with a Norm-Linear-GELU layer. To better assess each token's locality, we leverage both the local embedding and the context information within each quadrant. Specifically, we first split $\boldsymbol{x}_{\mathrm{s}}$ in the channel dimension to obtain the local feature $\boldsymbol{x}_{\mathrm{s}}^{\mathrm{local}}$ and the context feature $\boldsymbol{x}_{\mathrm{s}}^{\mathrm{global}}$:

$$\boldsymbol{x}_{\mathrm{s}}^{\mathrm{local}}, \ \boldsymbol{x}_{\mathrm{s}}^{\mathrm{global}} = \boldsymbol{x}_{\mathrm{s}}[0 : \frac{C}{2}], \ \boldsymbol{x}_{\mathrm{s}}[\frac{C}{2} : C], \quad \{\boldsymbol{x}_{\mathrm{s}}^{\mathrm{local}}, \boldsymbol{x}_{\mathrm{s}}^{\mathrm{global}}\} \in \mathbb{R}^{H \times W \times \frac{C}{2}}. \tag{5}$$

Then, we obtain $2 \times 2$ context vectors through an adaptive pooling layer and broadcast each context vector $\mathbf{v}_{\mathrm{s}}^{\mathrm{local}}$ into the local embedding along the channel dimension:

$$\begin{aligned} \boldsymbol{v}_{\mathrm{s}}^{\mathrm{local}} &= \mathrm{AdaptiveAvgPool2D}(\boldsymbol{x}_{\mathrm{s}}^{\mathrm{local}}), \quad \boldsymbol{v}_{\mathrm{s}}^{\mathrm{local}} \in \mathbb{R}^{2 \times 2 \times \frac{C}{2}}, \\ \boldsymbol{x}_{\mathrm{s}}^{\mathrm{agg}} &= \mathrm{Concat}(\boldsymbol{x}_{\mathrm{s}}^{\mathrm{global}}, \mathrm{Interpolate}(\boldsymbol{v}_{\mathrm{s}}^{\mathrm{local}})), \quad \boldsymbol{x}_{\mathrm{s}}^{\mathrm{agg}} \in \mathbb{R}^{H \times W \times C}, \end{aligned} \tag{6}$$

where Interpolate$(\cdot)$ is the bilinear interpolation operator that upsamples the context vector to a spatial size of $H \times W$. Thus, we obtain the aggregated score embedding $\mathbf{x}_s^{\text{agg}}$ and feed it to a Linear-GELU layer $\phi_p$ for partition score prediction:

$$s(i, j) = \text{Softmax}(\phi_p(\boldsymbol{x}_s^{\text{agg}})), \quad \boldsymbol{s} \in \mathbb{R}^{H \times W \times 2}, \tag{7}$$

where $\mathbf{s}(i, j)$ denotes the partition score of the token at spatial coordinate $(i, j)$.

**Quadtree-based window partition.** With each feature token's partition score $\mathbf{s}(i, j)$ predicted, we apply a quick quadtree-based window partition strategy with negligible computation cost. We construct bi-level window quadrants to capture spatial locality from coarse to fine. The quadtree-based strategy partitions the image feature into $N_1 = 4$ sub-windows at the coarse level and $N_2 = 16$ sub-windows at the fine level. Different from the Transformer scheme, which maintains the spatial shape of image features with arbitrary Key/Value features simply by setting the Query feature as the same size, the token sequence generated by the learned scan for QuadMamba should be merged into the feature with the original spatial size. Thus, we select the top $K = 1$ quadrant with the highest averaged local adjacency score at the coarse level and further partition it into four sub-windows:

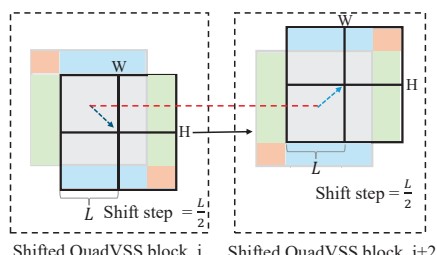

Figure 4: Omnidirectional window shifting scheme.

$$\begin{aligned}
\{\boldsymbol{w}_{N_1}^i | \, i = 0, 1, 2, 3\} &= \text{QuadWindowPartition}(\boldsymbol{x}), \\
\{\boldsymbol{s}_{N_1}^i | \, i = 0, 1, 2, 3\} &= \text{AdaptiveAvgPool2D}(\text{QuadWindowPartition}(\boldsymbol{s})) \\
\text{k} &\xleftarrow{\text{index}} \text{TopK}(\, \boldsymbol{s}_{N_1}^i \,|\, \text{K} = 1), \quad \text{i} \in \{0, 1, 2, 3\} \\
\{\boldsymbol{w}_{N_2}^{(\text{k,j})} | \, j = 0, 1, 2, 3\} &= \text{QuadWindowPartition}(\boldsymbol{w}_{N_1}^k),
\end{aligned} \tag{8}$$

where coarse-level windows $\boldsymbol{w}_{N_1}^i$ and fine-level sub-windows $\boldsymbol{w}_{N_1}^j$ have spatial sizes of $\{\frac{H}{2}, \frac{W}{2}\}$ and $\{\frac{H}{4}, \frac{W}{4}\}$, respectively. Afterward, we construct a 1D token sequence from the coarse-level windows $\{\boldsymbol{w}_{N_1}^i | \, i \neq k\}$ and fine-level sub-windows $\{\boldsymbol{w}_{N_2}^{(\text{k,j})} | \, j = 0, 1, 2, 3\}$ based on their relative spatial indices.

**Differentiable 1D sequence construction.** Although our target is to conduct adaptive learned window quadrant partition, it is non-trivial to construct the 1D token sequence from the 2D spatial windowed features. Directly sampling from the 2D feature according to the learned index for each sequence token is non-differentiable, which impedes end-to-end training. To overcome this, we apply a sequence masking strategy to formulate the token sequence, implemented by the Gumbel-Softmax technique [27]:

$$\mathbf{M}_{N_1} = \text{Gumbel-Softmax}(\{\boldsymbol{s}_{N_1}^i | \, i = 0, 1, 2, 3\}) \in \{0, 1\}, \mathbf{M}_{N_1} \in \mathbb{R}^{2 \times 2 \times 1}, \tag{9}$$

where value "1" in $\mathbf{M}_{N_1}$ represents the selected coarse-level windows, which will be divided into subwindows further; "0" denotes those non-selected. A differentiable operator, Gumbel-Softmax effectively gets around the non-differential index selection in Eqn. 8. Then, we first obtain two token sequences in the coarse and fine levels separately:

$$\begin{aligned}
\mathbf{L}_{N_1} &= \text{QuadWindowArrange}(\boldsymbol{x} \,|\, (\frac{H}{2}, \frac{W}{2})), \\
\mathbf{L}_{N_2} &= \text{QuadWindowArrange}(\text{Restore}(\mathbf{L}_{N_1}) \,|\, (\frac{H}{4}, \frac{W}{4})),
\end{aligned} \tag{10}$$

where QuadWindowArrange$(\cdot|(\text{h}, \text{w}))$ flattens the 2D feature into a 1D sequence based on the fixed size $(\text{h}, \text{w})$ window partition order. Restore$(\cdot)$ is to convert the 1D sequence back to the quadtree-window partitioned 2D feature. Next, we upsample the mask $\mathbf{M}_{N_1}$ to obtain the full sequence mask $\mathbf{M} \in \mathbb{R}^{H \times W \times 1}$. We then use Hadamard product $\odot$ and element-wise summation $\oplus$ to construct the 1D token sequence with negligible computation cost:

$$\mathbf{L} = (\mathbf{M} \odot \mathbf{L}_{N_2}) \oplus ((\mathbf{1} - \mathbf{M}) \odot \mathbf{L}_{N_1}), \tag{11}$$

where sequence **L** contains tokens in both coarse- and fine-level windows and is sent to the SS2D block for sequence modeling.

**Omnidirectional window shifting.** Considering the case that the most informative tokens are crossing two adjacent window quadrants, we borrow the idea of a shifted window scheme in Swin Transformer [42]. The difference is that the Transformer ignores the spatial locality for each token inside the window, while the token sequence inside the window in Mamba is still directional. Thus, we add additional shifted directions in the subsequent VSS blocks as shown in Fig 4, compared to only one direction shifting in Swin Transformer.

### 4.3 Model Configuration

It is noteworthy that QuadMamba's model capacity can be customized by tweaking the input feature dimension $d$ and the number of (Q)VSS layers $\{S_i\}_{i=1}^4$. In this work, we build four variants of the QuadMamba architecture, QuadMamba-Li/T/S/B, with varying capacities:

- Lite – block:$\{2, 2, 2, 2\}$, QuadVSS stages:$\{1, 2\}$, #Params: 5.4M, FLOPs: 0.82G.
- Tiny – block:$\{2, 6, 2, 2\}$, QuadVSS stages:$\{1, 2\}$, #Params: 10.3M, FLOPs: 2.0G.
- Small – block:$\{2, 2, 5, 2\}$, QuadVSS stages:$\{1, 2\}$, #Params: 31.2M, FLOPs: 5.5G.
- Base – block:$\{2, 2, 15, 2\}$, QuadVSS stages:$\{1, 2\}$, #Params: 50.6M, FLOPs: 9.3G.

In all these variants, QuadVSS blocks are placed in the specified QuadVSS model stages to bring into full play their locality preservation capabilities on higher-resolution features, Omnidirectional shifting layers are applied in every other QuadVSS blocks. More details are found in the Appendix.

## 5 Experiment

We conduct experiments on commonly used benchmarks, including ImageNet-1k [29] for image classification, MS COCO2017 [37] for object detection and instance segmentation, and ADE20K [78] for semantic segmentation. Our implementations follow prior works [42, 80, 41]. Detailed descriptions of the datasets and configurations are found in the Appendix. In what follows, we compare the proposed QuadMamba with mainstream vision backbones and conduct extensive ablation studies to back the motivation behind QuadMamba's designs.

### 5.1 Image Classification on ImageNet-1k

Tab. 1 demonstrates the superiority of QuadMamba in terms of accuracy and efficiency. Specifically, QuadMamba-S surpasses RegNetT-8G [50] by 2.5%, DeiT-S [58] by 2.6%, and Swin-T [42] by 1.1% Top-1 accuracy, with comparable or reduced FLOPs. This advantage holds when comparing other model variants of similar numbers of parameters or FLOPs. Compared to other Mamba-based vision backbones, QuadMamba also yields favorable performance under comparable network complexity. For instance, QuadMamba-B (83.8%) performs on par with or better than VMamba-B (83.7%), LocalVim-S (81.2%), and PlainMamba-L3 (82.3%), while having significantly less parameters and FLOPs. These results manifest the performance and complexity superiority of QuadMamba and its potential as a powerful yet highly efficient vision backbone. Furthermore, QuadMamba achieves similar or higher performance than LocalMamba while being completely free of the latter's expensive architecture and scan policy search, which makes it a more practical and versatile choice of vision backbone.

### 5.2 Object Detection and Instance Segmentation on COCO

On object detection and instance segmentation tasks, QuadMamba stands out as a highly efficient backbone among models and architectures of similar complexity, as measured by number of networks parameters and FLOPs. QuadMamba finds very few competitors under the category of tiny backbones with less than or around 30M parameters. As shown in Tab. 2, in addition to dramatically outperforming ResNet18 [20] and PVT-T [61], QuadMamba-T also leads EfficientVMamba-S [49] by considerable margins of 3.0% mAP on object detection and 2.1% on instance segmentation. Among larger backbones, QuadMamba once again surpasses all ConvNet-, Transformer-, and Mamba-based

Table 1: Image classification results on ImageNet-1k. Throughput (images / s) is measured on a single V100 GPU. All models are trained and evaluated on 224×224 resolution.

| | Model | #Params (M) | FLOPs (G) | Top-1 (%) | Top-5 (%) |
|---|---|---|---|---|---|
| ConvNet | ResNet-18 [20] | 11.7 | 1.8 | 69.7 | 89.1 |
| | ResNet-50 [20] | 25.6 | 4.1 | 79.0 | 94.4 |
| | ResNet-101 [20] | 44.7 | 7.9 | 80.3 | 95.2 |
| | RegNetY-4G [50] | 20.6 | 4.0 | 79.4 | 94.7 |
| | RegNetY-8G [50] | 39.2 | 8.0 | 79.9 | 94.9 |
| | RegNetY-16G [50] | 83.6 | 15.9 | 80.4 | 95.1 |
| Transformer | DeiT-S [58] | 22.1 | 4.6 | 79.8 | 94.9 |
| | DeiT-B [58] | 86.6 | 17.6 | 81.8 | 95.6 |
| | PVT-T [61] | 13.2 | 1.9 | 75.1 | 92.4 |
| | PVT-S [61] | 24.5 | 3.7 | 79.8 | 94.9 |
| | PVT-M [61] | 44.2 | 6.4 | 81.2 | 95.6 |
| | PVT-L [61] | 61.4 | 9.5 | 81.7 | 95.9 |
| | Swin-T [42] | 28.3 | 4.5 | 81.3 | 95.5 |
| | Swin-S [42] | 49.6 | 8.7 | 83.3 | 96.2 |
| | Swin-B [42] | 87.8 | 15.4 | 83.5 | 96.5 |
| Mamba | Vim-Ti [42] | 7 | – | 76.1 | 93.0 |
| | Vim-S [42] | 26 | – | 80.5 | 95.1 |
| | VMamaba-T [42] | 22 | 4.5 | 82.2 | – |
| | VMamaba-S [42] | 44 | 9.1 | 83.5 | – |
| | VMamaba-B [42] | 75 | 15.2 | 83.7 | – |
| | LocalVim-T [26] | 8 | 1.5 | 76.2 | – |
| | LocalVim-S [26] | 28 | 4.8 | 81.2 | – |
| | PlainMamba-L1 [66] | 7 | 3.0 | 77.9 | – |
| | PlainMamba-L2 [66] | 25 | 8.1 | 81.6 | – |
| | PlainMamba-L3 [66] | 50 | 14.4 | 82.3 | – |
| Ours | **QuadMamba-Li** | 5.4 | 0.8 | **74.2** | **92.1** |
| | **QuadMamba-T** | 10 | 2.0 | **78.2** | **94.3** |
| | **QuadMamba-S** | 31 | 5.5 | **82.4** | **95.6** |
| | **QuadMamba-B** | 50 | 9.3 | **83.8** | **96.7** |

rivals. Notably, QuadMamba-S outperforms EfficientVMamba-B, a Mamba-based backbone characterised by its higher efficiency, by 3.0% mAP on object detection and 2.2% on instance segmentation, using comparable parameters. Furthermore, QuadMamba-S is able to keep up with and even surpass the performance of LocalVMamba-T [26] while circumventing a whole load of architecture and scan search hassles not reflected by Tab. 2's complexity measurements. These results show how QuadMamba can serve as a strong and versatile vision backbone with a pragmatic trade-off among computational complexity, design costs, and performance.

## 5.3 Semantic Segmentation on ADE20K

As shown in Tab. 3, under comparable network complexity and efficiency, QuadMamba achieves significantly higher segmentation precision than ConvNet-based ResNet-50/101 [20] and ConvNeXt [43], Transformer-based DeiT [58] and Swin Transformer [42], as well as most Mamba-based architectures [80, 49, 66]. For instance, QuadMamba-S with 47.2% mIoU reports superior segmentation precision to Vim-S (44.9%), LocalVim-S (46.4%), EfficientVMamba-B (46.5%), and PlainMamba-L2 (46.8%), and competitive results to VMamba-T (47.3%). Compared with LocalMamba-S/B, QuadMamba-S/B trails by small margins, yet enjoying the advantage of not incurring extra network searching costs. It is worth noting that the LocalMamba is designed by the Neural Architecture Search (NAS) technology, which is data-dependent and lacks flexibility with other data modalities and data sources.

## 5.4 Ablation Studies

We conduct ablation studies to validate QuadMamba's design choices from various perspectives. QuadMamba-T is used for all experiments unless otherwise stated.

Table 2: Object detection and instance segmentation results on the COCO val2017 split using the Mask RCNN [19] framework.

| Backbones | #Params (M) | FLOPs (G) | $AP^{box}$ | $AP^{box}_{50}$ | $AP^{box}_{75}$ | $AP^{mask}$ | $AP^{mask}_{50}$ | $AP^{mask}_{75}$ |
|---|---|---|---|---|---|---|---|---|
| R18 [20] | 31 | 207 | 34.0 | 54.0 | 36.7 | 31.2 | 51.0 | 32.7 |
| PVT-T [61] | 32 | 208 | 36.7 | 59.2 | 39.3 | 35.1 | 56.7 | 37.3 |
| ViL-T [71] | 26 | 145 | 41.4 | 63.5 | 45.0 | 38.1 | 60.3 | 40.8 |
| EfficientVMamba-S [49] | 31 | 197 | 39.3 | 61.8 | 42.6 | 36.7 | 58.9 | 39.2 |
| **QuadMamba-Li** | 25 | 186 | **39.3** | **61.7** | **42.4** | **36.9** | **58.8** | **39.4** |
| **QuadMamba-T** | 30 | 213 | **42.3** | **64.6** | **46.2** | **38.8** | **61.6** | **41.4** |
| R50 [20] | 44 | 260 | 38.6 | 59.5 | 42.1 | 35.2 | 56.3 | 37.5 |
| PVT-S [61] | 44 | - | 40.4 | 62.9 | 43.8 | 37.8 | 60.1 | 40.3 |
| Swin-T [39] | 48 | 267 | 42.7 | 65.2 | 46.8 | 39.3 | 62.2 | 42.2 |
| ConvNeXt-T [43] | 48 | 262 | 44.2 | 66.6 | 48.3 | 40.1 | 63.3 | 42.8 |
| EfficientVMamba-B [49] | 53 | 252 | 43.7 | 66.2 | 47.9 | 40.2 | 63.3 | 42.9 |
| PlainMamba-L2 [66] | 53 | 542 | 46.0 | 66.9 | 50.1 | 40.6 | 63.8 | 43.6 |
| ViL-S [71] | 45 | 218 | 44.9 | 67.1 | 49.3 | 41.0 | 64.2 | 44.1 |
| VMamba-T [41] | 42 | 262 | 46.5 | 68.5 | 50.7 | 42.1 | 65.5 | 45.3 |
| LocalVMamba-T [26] | 45 | 291 | 46.7 | 68.7 | 50.8 | 42.2 | 65.7 | 45.5 |
| **QuadMamba-S** | 55 | 301 | **46.7** | **69.0** | **51.3** | **42.4** | **65.9** | **45.6** |

Table 3: Semantic segmentation results on ADE20K using UperNet [62]. mIoUs are measured with single-scale (SS) and multi-scale (MS) testings on the *val* set. FLOPs are measured with an input size of $512 \times 2048$.

| Backbone | Image size | #Params (M) | FLOPs (G) | mIoU (SS) | mIoU (MS) |
|---|---|---|---|---|---|
| EfficientVMamba-T [49] | $512^2$ | 14 | 230 | 38.9 | 39.3 |
| DeiT-Ti [58] | $512^2$ | 11 | - | 39.2 | - |
| Vim-Ti [80] | $512^2$ | 13 | - | 40.2 | - |
| EfficientVMamba-S [49] | $512^2$ | 29 | 505 | 41.5 | 42.1 |
| LocalVim-T [26] | $512^2$ | 36 | 181 | 43.4 | 44.4 |
| PlainMamba-L1 [66] | $512^2$ | 35 | 174 | 44.1 | - |
| **QuadMamba-T** | $512^2$ | 40 | 886 | **44.3** | **45.1** |
| ResNet-50 [20] | $512^2$ | 67 | 953 | 42.1 | 42.8 |
| DeiT-S + MLN [58] | $512^2$ | 58 | 1217 | 43.8 | 45.1 |
| Swin-T [42] | $512^2$ | 60 | 945 | 44.4 | 45.8 |
| Vim-S [80] | $512^2$ | 46 | - | 44.9 | - |
| LocalVim-S [26] | $512^2$ | 58 | 297 | 46.4 | 47.5 |
| EfficientVMamba-B [49] | $512^2$ | 65 | 930 | 46.5 | 47.3 |
| PlainMamba-L2 [66] | $512^2$ | 55 | 285 | 46.8 | - |
| VMamba-T [41] | $512^2$ | 55 | 964 | 47.3 | 48.3 |
| LocalVMamba-T [26] | $512^2$ | 57 | 970 | 47.9 | 49.1 |
| **QuadMamba-S** | $512^2$ | 62 | 961 | **47.2** | **48.1** |
| ResNet-101 [20] | $512^2$ | 85 | 1030 | 42.9 | 44.0 |
| DeiT-B + MLN [58] | $512^2$ | 144 | 2007 | 45.5 | 47.2 |
| Swin-S [42] | $512^2$ | 81 | 1039 | 47.6 | 49.5 |
| PlainMamba-L3 [66] | $512^2$ | 81 | 419 | 49.1 | - |
| VMamba-S [41] | $512^2$ | 76 | 1081 | 49.5 | 50.5 |
| LocalVMamba-S [26] | $512^2$ | 81 | 1095 | 50.0 | 51.0 |
| **QuadMamba-B** | $512^2$ | 82 | 1042 | **49.7** | **50.8** |

**Effect of locality in Mamba.** We factorise the effect of coarse- and fine-grain locality modeling in building 1D token sequences on model performance. Specifically, we compare the naive window-free flattening strategy in [80, 41] that overlooks 2D locality against window partitions in three scales (i.e., $28 \times 28$, $14 \times 14$, $2 \times 2$) that represent three granularity levels of feature locality. In practice, we replace QuadVSS blocks of a QuadMamba-T model with the plain VSS blocks in [41]. To exclude the negative effects of padding operations, we only partition the features with a spatial size of $56 \times 56$ in the first model stage. As illustrated in Tab. 4, the naive scan strategy leads to significantly degraded object detection and instance segmentation performance compared to when windowed scan is adopted. The scale of local windows is also shown to considerably influence the model performance, which suggests that too large or too small a window given the image resolution can be suboptimal.

**Quadtree-based partition resolutions.** We examine the choice of partition resolutions in the bi-level quadtree-based partition strategy. The configured resolutions in Tab. 5 are applied in the first two

Table 4: Impact of using different local window sizes and the naive flattening strategy.

| Window Size | Top-1 (%) | $AP^{box}$ | $AP^{mask}$ |
|---|---|---|---|
| w/o windows | 72.2 | 33.1 | 30.5 |
| $28 \times 28$ | 72.9 | 33.8 | 31.7 |
| $14 \times 14$ | **73.5** | **35.8** | **32.1** |
| $2 \times 2$ | 72.4 | 33.4 | 30.6 |

Table 5: Impact of different local window partition resolutions.

| Coarse | Fine | Top-1 (%) |
|---|---|---|
| $(H, W)$ | $(\frac{H}{2}, \frac{W}{2})$ | 73.5 |
| $(\frac{H}{2}, \frac{W}{2})$ | $(\frac{H}{4}, \frac{W}{4})$ | **73.8** |
| $(\frac{H}{4}, \frac{W}{4})$ | $(\frac{H}{8}, \frac{W}{8})$ | 73.6 |

Table 6: Impact of different number of (Quad)VSS blocks per stage.

| Depth | #Params. | FLOPs | Top-1 (%) |
|---|---|---|---|
| 2-2-8-2 | 31 | 9.2 | 82.0 |
| 2-8-2-2 | 38 | 8.1 | 81.5 |
| 2-2-2-8 | 74 | 7.8 | 81.8 |
| 2-4-6-2 | 36 | 8.5 | **82.1** |

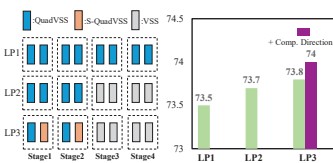

Figure 5: Impact of different layer patterns and shift directions.

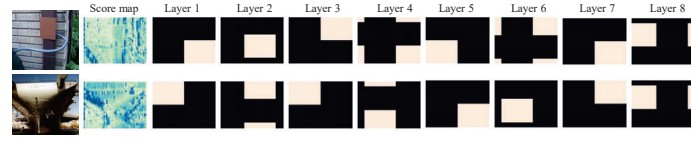

Figure 6: Visualization of partition maps that focus on different regions from shallow to deep blocks.

model stages with feature resolutions of $\{56 \times 56, 28 \times 28, 14 \times 14\}$. Experimentally we deduce the optimal resolutions to be $\{1/2, 1/4\}$ for the coarse- and fine-level window partition, respectively. This handcrafted configuration may be replaced by more flexible and learnable ones in future work.

**Layer patterns in the hierarchical model stage.** We investigate different design choices of the layer pattern within our hierarchical model pipeline. From Fig. 5, layer pattern LP2 outperforms LP1 with less QuadVSS blocks by $0.2\%$ accuracy. This is potentially due to the effect of locality modeling being more pronounced in shallower stages than in deeper stages as well as the adverse influence of padding operations in stage 3. LP3, which places the QuadVSS blocks in the first two stages and in an interleaved manner, achieves the best performance, and is adopted as our model design.

**Necessity of multi-directional window shift.** Different from the unidirectional shift in Swin Transformer [42], Fig. 5 shows a $0.2\%$ gain in accuracy as complementary shifting directions are added. This is expected since attention in Transformer is non-causal, whereas 1D sequences in Mamba, being causal in nature, are highly sensitive to relative position. A multi-directional shifting operation is also imperative for handling cases where the informative region spans across adjacent windows. Fig. 6 further visualizes the shifts in the learned fine-grained quadrants across hierarchical stages, which adaptively attend to different spatial details at different layers.

**Numbers of (Quad)VSS blocks per stage.** We conduct experiments to evaluate the impact of different numbers of (Quad)VSS blocks in each stage. Tab. 6 presents four configurations, following design rule LP3 in Fig. 5, with a fixed channel dimension of 96. We find that a bulky 2nd or 4th stage leads to diminished performance compared to a heavy 3rd stage design, whereas dealing out the (Quad)VSS blocks more evenly between stages 2 and 3 yields comparable if not better performance with favorable complexities. This evidence can serve as a rule of thumb for model design in future work, especially as the model scales up.

# 6 Conclusion

In this paper, we propose QuadMamba, a vision Mamba architecture that serves as a versatile and efficient backbone for visual tasks, such as image classification and dense predictions. QuadMamba effectively captures local dependencies of different granularities by learnable quadtree-based scanning, which adaptively preserves the inherent locality within image data with negligible computational overheads. The QuadMamba's effectiveness has been proven through extensive experiments and ablation studies, outperforming popular CNNs and vision transformers. However, one limitation of QuadMamba is that window partition with more than two levels is yet to be explored, which may be particularly relevant for handling dense prediction visual tasks and higher-resolution data, such as remote sensing images. The fine-grained partition regions are rigid and lack flexibility in attending to regions of arbitrary shapes and sizes, which is left for our future work. We hope that our approach will motivate further research in applying Mamba to more diverse and complex visual tasks.

**Acknowledgments.** This work was supported by NSFC (62322113, 62376156), Shanghai Municipal Science and Technology Major Project (2021SHZDZX0102), and the Fundamental Research Funds for the Central Universities.

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

# A  Appendix

In the supplementary materials, we provide more details and analysis of QuadMamba in Sec. A.1, implementation details in Sec. A.2, and more information on QuadMamba's model variants in Sec. A.3. In Sec.A.4, we also provide the pseudo-code to help understand the key operations within the QuadVSS block. Next, we present the complexity analysis and the throughput measurement of our proposed QuadMamba variants in Sec.A.5. Finally, we provide more visualization results of QuadMamba in Sec.A.6.

## A.1  Schematic Illustration

To better assess the impact of the Mamba model and our proposed learnable scanning method, we integrated our key token operator into the meta-architecture of the Vision Transformer, as illustrated in Fig. 7.

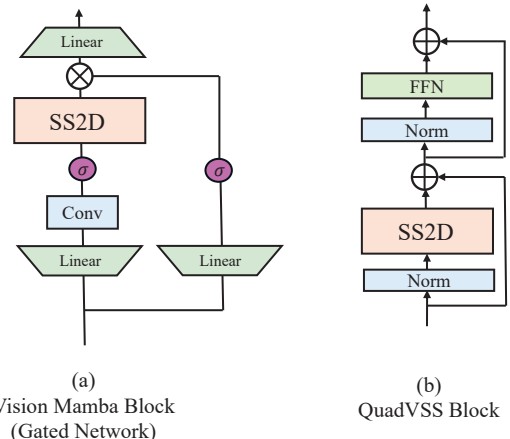

| (a) | (b) |
| Vision Mamba Block | QuadVSS Block |
| (Gated Network) | |

Figure 7: Architecture of (a) the vanilla VSS block, which is in the form of a gated network, and (b) our modified block, which adopts the meta-architecture of the transformer and comprises a token operator, an FFN, and residual connections.

**Relationship to RNNs**. The concept of recurrence in a hidden state is closely linked to Recurrent Neural Networks (RNNs) and State Space Models (SSMs), as suggested by [13]. Some RNN variants employ forms of gated RNNs and remove the time-wise nonlinearities. These can be considered as a combination of the gating mechanisms and selection mechanisms.

## A.2  Implementation Details

Following the VMamba [41], we conducted a benchmark to assess the image classification performance of QuadMamba on the ImageNet-1k [29] dataset. ImageNet [29] is widely recognized as the standard for image classification benchmarks, consisting of around 1.3 million training images and 50,000 validation images spread across 1,000 classes.

The training scheme is based on DeiT [58]. The data augmentation techniques used include random resized crop (input image size of 2242), horizontal flip, RandAugment [77], Mixup [70], CutMix [69], Random Erasing [77], and color jitter. Additionally, regularization techniques such as weight decay, stochastic depth [24], and label smoothing [56] are applied. All models are trained using AdamW [45]. The learning rate scaling rule is calculated as $\frac{BatchSize}{1024} \times 10^{-3}$. Our models are implemented with PyTorch and Timm libraries and trained on A800 GPUs.

For object detection and instance segmentation, we assess QuadMamba using the MS COCO2017 dataset [37]. The MS COCO2017 dataset is widely used for object detection and instance segmentation and consists of 118,000 training images, 5,000 validation images, and 20,000 test images of common objects.

Table 7: Our detailed model variants for ImageNet-1k. Here, The definitions are as follows: "$\mathrm{Conv} - k\_c\_s$" denotes convolution layers with kernel size $k$, output channel $c$ and stride $s$. "$\mathrm{MLP}\_c$" is the MLP structure with hidden channel $4c$ and output channel $c$. And "$(\mathrm{Quad})\mathrm{VSS}\_n\_r$" is the VSS operation with the dimension expansion ratio $n$ and the channel dimension $r$. "C" is 48 for QuadMamba-Li and 64 for QuadMamba-S, and 96 for QuadMamba-B and QuadMamba-L.

| Name | Output | Lite | Small | Base | Large |
|---|---|---|---|---|---|
| stem | $56 \times 56$ | \multicolumn{4}{c}{patch_embed: Conv-3_C/2_2, Conv-3_C/2_1, Conv-3_C_2} |
| stage1 | $56 \times 56$ | $\begin{bmatrix} \text{QuadVSS\_1\_48} \\ \text{MLP\_48} \end{bmatrix} \times 2$ | $\begin{bmatrix} \text{QuadVSS\_1\_64} \\ \text{MLP\_64} \end{bmatrix} \times 2$ | $\begin{bmatrix} \text{QuadVSS\_2\_96} \\ \text{MLP\_96} \end{bmatrix} \times 2$ | $\begin{bmatrix} \text{QuadVSS\_2\_96} \\ \text{MLP\_96} \end{bmatrix} \times 2$ |
| | | \multicolumn{4}{c}{patch_embed: Conv-3_2C_2} |
| stage2 | $28 \times 28$ | $\begin{bmatrix} \text{QuadVSS\_1\_96} \\ \text{MLP\_96} \end{bmatrix} \times 2$ | $\begin{bmatrix} \text{QuadVSS\_1\_128} \\ \text{MLP\_128} \end{bmatrix} \times 6$ | $\begin{bmatrix} \text{QuadVSS\_2\_192} \\ \text{MLP\_192} \end{bmatrix} \times 2$ | $\begin{bmatrix} \text{QuadVSS\_2\_192} \\ \text{MLP\_192} \end{bmatrix} \times 2$ |
| | | \multicolumn{4}{c}{patch_embed: Conv-3_4C_2} |
| stage3 | $14 \times 14$ | $\begin{bmatrix} \text{VSS\_1\_192} \\ \text{MLP\_192} \end{bmatrix} \times 2$ | $\begin{bmatrix} \text{VSS\_1\_256} \\ \text{MLP\_256} \end{bmatrix} \times 5$ | $\begin{bmatrix} \text{VSS\_2\_384} \\ \text{MLP\_384} \end{bmatrix} \times 5$ | $\begin{bmatrix} \text{VSS\_2\_384} \\ \text{MLP\_384} \end{bmatrix} \times 15$ |
| | | \multicolumn{4}{c}{patch_embed: Conv-3_8C_2} |
| stage4 | $7 \times 7$ | $\begin{bmatrix} \text{VSS\_1\_384} \\ \text{MLP\_384} \end{bmatrix} \times 2$ | $\begin{bmatrix} \text{VSS\_1\_512} \\ \text{MLP\_512} \end{bmatrix} \times 2$ | $\begin{bmatrix} \text{VSS\_2\_768} \\ \text{MLP\_768} \end{bmatrix} \times 2$ | $\begin{bmatrix} \text{VSS\_2\_768} \\ \text{MLP\_768} \end{bmatrix} \times 2$ |
| Classifier | $1 \times 1$ | \multicolumn{4}{c}{average pool, 1000d fully-connected} |
| GFLOPs | | 0.82G | 2.07G | 5.51G | 9.30G |
| Params | | 5.47M | 10.32M | 31.25M | 50.6M |

Table 8: Increased model costs when QuadVSS blocks are applied in the first two stages. The blocks in four stages are $(2, 2, 2, 2)$.

| QuadBlock | Channel | #Params. | FLOPs |
|---|---|---|---|
| ✗ | 48 | 5.4 | 0.78 |
| ✓ | 48 | 5.5 (+1.8%) | 0.86 (+10.2%) |
| ✗ | 96 | 24.8 | 3.7 |
| ✓ | 96 | 27.1 (+9.2%) | 4.8 (+31.6%) |

For semantic segmentation, we evaluate QuadMamba on the ADE20K [78] dataset. ADE20K is a popular benchmark for semantic segmentation with 150 categories. It consists of 20,000 training images and 2,000 validation images. Following Swin Transformer [42], we construct UperHead [62] with the pretrained model. The learning rate is set as $6 \times 10^{-5}$. The fine-tuning process consists of a total of 160,000 iterations with a batch size of 16. The default input resolution is 512x512, and the experimental results are using 640x640 inputs and multi-scale (MS) testing.

## A.3 Model Variants

We present the detailed architectural configurations of QuadMamba variants in Tab. 7. Each building unit of the model variants and the corresponding hyper-parameters are illustrated in detail. We mostly place the proposed QuadVSS block in the first two stages and place the plain VSS block from [41] in the latter two stages. The reason is that the features in the shallow stage have larger spatial resolution and need more locality modeling.

## A.4 Key Operators in QuadVSS Block

In this section, we present a detailed illustration of the QuadVSS block. The overall structure of the QuadVSS block is presented in Alg. 1. The process to partition feature tokens via the bi-level quadtree-based strategy is outlined in Alg. 2, and that to restore them back into a 2D spatial feature is described in Alg. 3. Moreover, in Alg. 4, we provide the pseudo-code of the proposed sequence masking strategy to construct a 1D token sequence in a differential manner. In Fig. 8, we illustrate the detailed pipeline of the ominidirectional shifting scheme in two QuadVSS blocks. Moreover, in

Table 9: Throughputs of QuadMamba variants. Measurements are taken with an A800 GPU.

| Model | #Params. | FLOPs | Throughput |
|---|---|---|---|
| QuadMamba-Li | 5.4 | 0.8 | 1754 |
| QuadMamba-T | 10.3 | 2.1 | 1586 |
| QuadMamba-S | 31.2 | 5.5 | 1252 |
| QuadMamba-B | 50.6 | 9.3 | 582 |

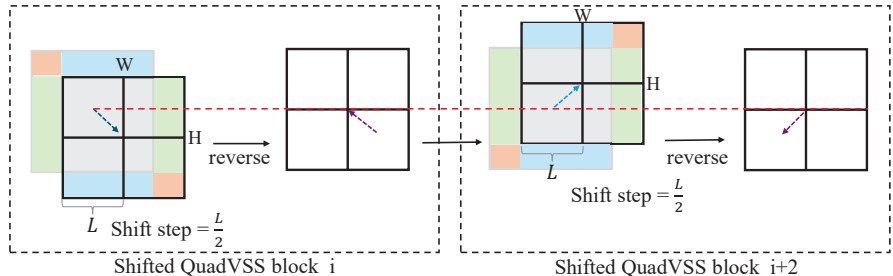

Figure 8: Ominidirectional shifting in two successive QuadVSS blocks. Two directions are complementary to each other, which mitigates the issue of the informative region spanning across adjacent windows.

Fig. 9, we provide the details of three different quadtree-based partition resolution configurations at coarse and fine levels, which are mentioned in Tab. 5.

## A.5 Complexity & Throughput Analysis

Our learnable scanning method does not alter the total sequence length $L = H \times W$. Thus, the QuadVSS block retains the linear computational complexity of $\mathcal{O}(L)$ of mainstream vision Mamba architectures [80, 41, 26, 66]. Other additional model costs are negligible. In Tab. 9, we show the throughput of the proposed QuadMamba variants, which are measured in the V800 GPU platform.

The computation complexity of a standard Transformer is as follows: Assuming its input $X \in \mathbb{R}^{N \times D}$ has a total input token number of $N = H \times W$ and channel dimensions of $D$, the FLOPs for the transformer attention can be calculated as:

$$\text{FLOPs} = 4HWD^2 + 2(HW)^2D = \mathcal{O}(N^2). \tag{12}$$

It shows a quadratic complexity with the input size of the transformer attention scheme, as compared to the linear complexity of QuadMamba.

## A.6 Visualization Results

In this section, we provide more visualization results. In Fig. 10, we visualize the partition maps that focus on different regions from shallow to deep blocks, using QuadMamba-T as an example. In Fig. 11, we showcase more visualizations of the Effective Receptive Field (ERF) of CNN, Transformer, and Mamba variants. As can be observed, QuadMamba exhibits not only a global ERF but also a more locality-sensitive response compared to VMamba [41] with a naive flattening strategy.

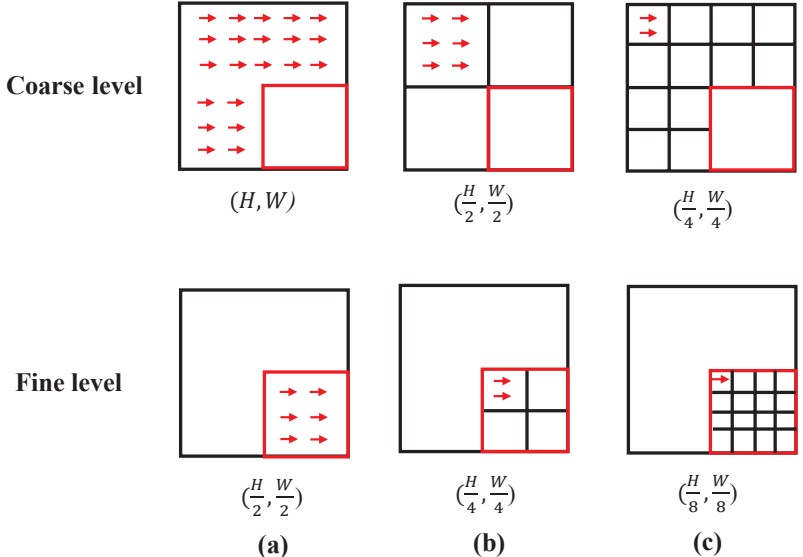

Figure 9: Details of the three different local window partition resolution configurations.

---

**Algorithm 1** PyTorch code of QuadVSS block

---

```python
import torch
import torch.nn as nn

class QuadVSSBlock(nn.Module):
    def __init__(self, dim, expension_ratio=8/3, sssm_d_state: int=16, conv_ratio=1.0,
        ssm_dt_rank:Any ="auto", ssm_conv:int=3, ssm_conv_bias=True, ssm_drop_rate:float
        =0, mlp_ratio = 4.0, mlp_act_layer = nn.GELU, norm_layer=partial(nn.LayerNorm,
        eps=1e-6), act_layer=nn.GELU, drop_path=0.):
        super().__init__()
        self.norm1 = norm_layer(dim)
        self.norm2 = norm_layer(dim)
        hidden = int(expension_ratio * dim)
        self.token_op = QuadSS2D(d_model = hidden, d_state = ssm_d_state, ssm_ratio =
            ssm_ratio, dt_rank = ssm_dt_ranl, act_layer = ssm_act_layer, d_conv =
            ssm_conv, conv_bias = ssm_conv_bias, dropout = ssm_drop_rate )
        self.drop_path = DropPath(drop_path)
        self.mlp = MLP(hidden, drop = mlp_drop_rate)

    def forward(self, x):
        x = input
        x = x + self.drop_path(self.norm1(self.token_op(x)))
        x = x + self.drop_path(self.norm2(self.mlp(x)))
        return x
```

---

**Algorithm 2** PyTorch code of Quadtree window partition at two levels

```
import torch
import torch.nn as nn

def coarse_level_quadtree_window_partition(x, H, W, quad=2)
    B, C, L = x.shape
    x = x.view(B, C, H, W)
    quad1 = quad2 = quad
    h = math.ceil(H / quad)
    w = math.ceil(w /quad)
    x= x.view(B,c,quad1,h, quad2,w).permute(0,1,2,4,3,5).reshape(B, c, -1)
    return x

def fine_level_quadtree_window_partition(x, H, W, quad=2)
    B, C, L = x.shape
    x = x.view(B, C, H, W)
    quad1 = quad2 = quad3 = quad4 = quad
    h = math.ceil((H / quad) / quad)
    w = math.ceil((W / quad) / quad)
    x= x.view(B, c, quad1, quad3*h, quad2, quad4*w).view(B, C, quad1, quad3, h, quad2,
        quad4, w).permute(0, 1, 2, 3, 5, 6, 4, 7).reshape(B, C, -1)
    return x
```

**Algorithm 3** PyTorch code of Quadtree window restoration at two levels

```
def coarse_level_quadtree_window_restoration(y, H, W, quad=2)
    B, C, L = y.shape
    quad1 = quad2 = quad
    h = math.ceil(H / quad)
    w = math.ceil(w /quad)
    y= y.view(B,c,quad1,quad2, h,w).permute(0, 1, 2, 4, 3, 5).reshape(B, c, -1)
    return x

def fine_level_quadtree_window_restoration(y, H, W, quad=2)
    B, C, L = y.shape
    quad1 = quad2 = quad3 = quad4 = quad
    h = math.ceil((H / quad) / quad)
    w = math.ceil((W / quad) / quad)
    y= y.view(B, c, quad1, quad3, quad2, quad4, h, w).permute(0, 1, 2, 3, 6, 4, 5, 7).
        reshape(B, C, -1)
    return y
```

**Algorithm 4** PyTorch code of differentiable sequence masking

```
x_rs = x.reshape(B, D, -1)
score_window = F.adaptive_avg_pool2d(score[:, 0:1, :, :], (2, 2)) # b, 1, 2, 2
hard_keep_decision = F.gumbel_softmax(score_window.view(B, 1, -1), dim=-1, hard=True).
    unsqueeze(-1).unsqueeze(-1) #[b, 1, 4, 1, 1]

hard_keep_decision_mask = window_expansion(hard_keep_decision, H=int(H), W=int(W)) #[b,
    1, l]
x_masked_select = x_rs * hard_keep_decision_mask
x_masked_nonselect = x_rs * (1.0 - hard_keep_decision_mask)
# local scan quad region
x_masked_select_localscan = local_scan_quad_quad(x_masked_select, H=H, W=W) #BCHW -> B,C
    ,L
x_masked_nonselect_localscan = local_scan_quad(x_masked_nonselect, H=H, W=W) #BCHW -> B,
    C,L
x_quad_window = x_masked_nonselect_localscan + x_masked_select_localscan# B,C,L
```

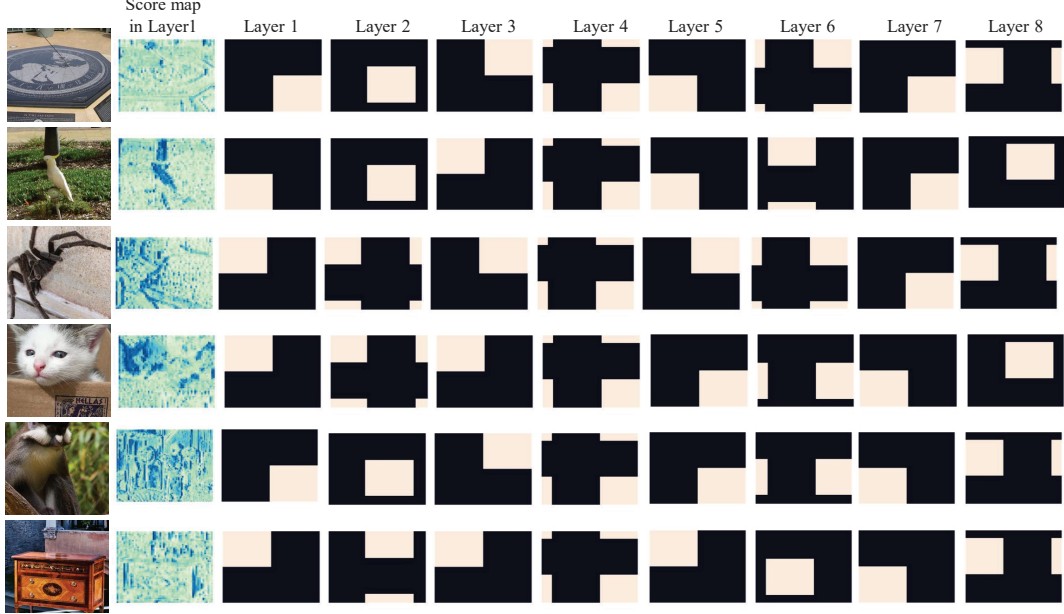

Figure 10: Visualization of partition maps which focus on different regions from shallow to deep blocks. The second column shows the partition score maps in the $1^{st}$ block.

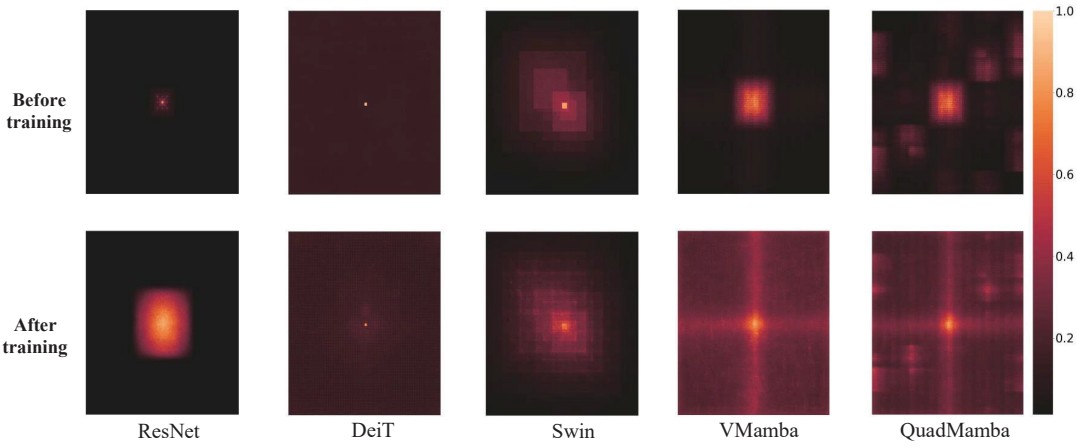

Figure 11: Visualization of Effective Receptive Field (ERF) of CNN, Transformer, and Mamba variants. Our QuadMamba exhibits not only a global ERF but also more locality-sensitive response compared to VMamba with a naive flattening strategy.

