# OpenReview forum: "QuadMamba: Learning Quadtree-based Selective Scan for Visual State Space Model"
_NeurIPS.cc/2024/Conference — NeurIPS 2024 poster_

### Official Review · Reviewer_VDbd · 2024-07-07

**Soundness:** 4
**Presentation:** 4
**Contribution:** 4
**Rating:** 9
**Confidence:** 4

**Summary:**

This paper presents QuadMamba, a novel Mamba architecture for visual tasks such as image classification and dense predictions. Unlike the classic Vision Mamba, which splits 2D visual data using fixed windows, the authors introduce learnable windows by using a lightweight module that predicts 2D locality more informatively, allowing for the ignoring of irrelevant windows and further splitting of the most informative windows into sub-windows, capturing more fine-grained information. This coarse-to-fine fashion is made possible by the new implementation of a splitting operator with Hadamard product and element-wise summation, making the pipeline fully differentiable. The authors experiment on the classic benchmarks for image classification (ImageNet), object detection (COCO), and semantic segmentation (ADE20K), showing that the method achieves state-of-the-art results.

**Strengths:**

- First, the paper is very well written and most of the figures are done very well so that it is easy to fully understand the story of paper.

- I also like the idea coarse-to-fine scanning, to capture more relevant information in each layer in QuadMamba. It is also worth noting that as the authors mentioned, direct sampling from 2D visual data based on index is not differentiable, but the authors has proposed a solution to overcome it which I consider is a strong contribution.

- The experiment shows that QuadMamba always outperforms the classic CNN, Vision Transformer on popular tasks (image classification, object detection, image segmentation).

**Weaknesses:**

I do not see any major weakness for this paper but only have a few questions for better understanding the paper and improve its clarify:

- The lightweight module to predict 2D locality, informative windows, is it shared across layers or specific to each layer? It seems this module is shared across layers but I feel it needs to be different as each layer capture different information.

- As the authors mentioned in limitation, in remote sensing images which might be helpful to have more than two levels partition in QuadMamba. However, I feel this experiment can still be experimented with the current architecture. In this case, what can happen is that the lightweight module will split into sub-quadrants for all windows ( since all of them are mostly capturing relevant for predicting the output). Did the authors observe this  extreme case to subplot all quadrants into sub-quadrants can happen with the current architecture?

**Questions:**

As mentioned in weakness section, I have only two questions related to lightweight module, extreme case of splitting all quadrants to sub-quadrants in each layer. Besides, it would be great if the authors can provide some experiments with more than two levels partition as mentioned in conclusion, for example, by using the lite model on semantic segmentation tasks should be enough. I would be very helpful to get some insights for possible future works.

**Limitations:**

Yes, the authors explicit mentioned the limitation which is that the partition with more than two levels is yet to be explored.

---

> ### Author Rebuttal · Authors · 2024-08-05
>
> ### Q1.Details about the lightweight prediction module
>
> Thanks for your valuable advice. We will depict the prediction module more clearly in the revised manuscript. In our design, each QuadVSS block has its specific prediction module that determines the informative regions of the current layer. The potential reason is that each layer needs to attend to different regions as the feature resolution and network depth change, given that features of different depths and resolutions are known to have different responce patterns in terms of regions and context. The shift scheme also helps the model select the informative regions more flexibly, which can enhance the feature representation quality.
>
> ### Q2.The effects of more window-level partitions
>
> Thanks for your thoughtful question. The downstream tasks (detection and segmentation) need the pre-training weight of the ImageNet classification. Our two-level window partition strategy mainly considers the input image resolution (224x224) of the image classification task. The input image resolution in dense prediction tasks is generally higher than that in the image classification task. Thus, our module design may not be optimal for downstream dense prediction tasks involving very high resolutions. We will further improve this window partition scheme in future work. A similar phenomenon is observed in Table 4 of the main text: A moderate local window size (14x14) perform better than overly-small (2x2) and overly-large (28x28) window sizes.
>
> Moreover, we conduct a simple experiment in semantic segmentation to explore the proposed question. We split the image into sub-quadrants for all windows using fixed local window sizes.  The size is increased four times in the table below.  It is worth noting that semantic segmentation results are highly related to the pre-training weights from the ImageNet Classification. Since we do not train each model variant on ImageNet classification, the results may not truly reflect the effects of window size in the segmentation experiment.  In the table below, the 8x8 window setting outperforms the 1x1 setting. This indicates that more window partitions can improve the segmentation performance in high-resolution inputs. It is hard to find the optimal local window size for a specific downstream task. The quadtree variant without pertaining achieves the results in between the 8x8 setting and 32x32 setting. It indicates that the two-level quadtree window partition setting may not be flexible enough to handle the high-resolution downstream task.  It is because we designed our hyper-parameter by mainly considering the image classification task (lower image resolution).  Thus, we will be devoted to designing more flexible window partition schemes to handle various downstream tasks in the future.
>
>
> | Model | Pre-training| Local Window | mIoU |
> |:------|:-------:|:-------:|------:|
> | Tiny (B2) | Yes|Quadtree | 44.3 |
> | Tiny (B2) | None|Quadtree | 38.0 |
> | Tiny (B2) | None|w/o (1x1) | 36.8 |
> |  Tiny (B2)| None|8x8 | 38. 2|
> |  Tiny (B2)| None|32x32 | 37.9 |
> |  Tiny (B2)| None|128x128 | 36. 9|

---

### Official Review · Reviewer_ALic · 2024-07-10

**Soundness:** 3
**Presentation:** 3
**Contribution:** 4
**Rating:** 8
**Confidence:** 4

**Summary:**

This paper proposes a vision mamba backbone for various vision tasks. It aims to adapt the recent popular mamba model originated from the language domain to vision tasks. The authors propose a learnable quad-tree partition strategy which can adaptively generate multi-scale visual sequences with spatial prior for mamba model. The authors also make contributions on differentiable training and shifted token interactions. Experimental results show the effectiveness of the proposed vision mamba, including image classification, object detection and segmentation. It shows the potential to be applied in various vision mamba models. Ablation studies are also extensively conducted.

**Strengths:**

- Strong Motivation. This paper is well-motivated and important. Effectively adapting the Mamba model to vision tasks has gained wide attention recently. Multi-grained image information and 2D priors are essential to building a vision backbone, which has led to a strong motivation for this paper. Treating the 2D image as token sequences in the language domain needs exploration and adaptation.

- Novel \& applicable method.  The quadtree-based window partition is novel in vision Mamba. The learnable module part is lightweight and conceptually simple. The quadtree search technique is seen in other tasks, such as the point cloud domain. This paper only learns to partition the semantic-rich content with a handcrafted partition strategy, which avoids developing a search algorithm that is too complex. The other two points, the differentiable trick and shifted scheme, are also interesting. It has the potential to be applied to constructing token sequences for other vision mamba models.

- Good experimental results. The method proposed is tested in image classification, detection, and segmentation tasks, which proves its effectiveness over other vision mamba models. The ablation studies and visualization results also give more explanations about the quadtree-based sequence construction.

- Well-written presentation.  The writing is clear and precise. In general, the presentation of this work is easy to follow.

The experiments are well performed and the presented method compares favorably to the considered baselines.

**Weaknesses:**

I have a number of suggestions and questions that may help to further improve the paper:

- It would be helpful to demonstrate the impact of the hyper-parameters in the model design. For example, is it helpful to partition more window levels?

- More illustrations and analysis are needed for the learnable parameters. It can show the relationship between model cost and performance gain to other researchers/readers.

- It would be useful to give the details about the training settings and compare them to other vision mamba works.

**Questions:**

I have listed some questions in the previous section.

**Limitations:**

Yes

---

> ### Author Rebuttal · Authors · 2024-08-05
>
> ### Q1.More illustrations of hyper-parameters
> Thanks for your valuable advice. We will add more analysis on the hyper-parameters. The impacts of the increased number of window partition levels are explored in Table 5 of the main text. We examine the choice of partition resolutions in the bi-level quadtree-based partition strategy. Experimentally we deduce the optimal resolutions to be {1/2, 1/4} for the coarse- and fine-level window partition, respectively. It is worth noting that the design space is handcrafted and may be extended to more levels. We will explore a more flexible scheme in the future.
>
> ### Q2.More analysis of learnable parameters
>
> Thanks for your insightful opinion. We will include more analysis on the learnable parameters in the revised manuscript. To show the relationship between model complexity and performance, we plot the performance against flops for different methods and models in Figure 2 of the attached PDF file. We also conduct an ablation experiment on the local and global context embedding for the prediction module. The method of combining both the local and global contexts outperforms the one using only the global context vector.
>
> | Variant| Embedding | Top-1 Acc.(%) |
> |:------|:-------:|------:|
> | A | Global | 74.0 |
> | B | Global+Local| 74.2 |
>
> ### Q3.More details about training settings
>
> We will add more illustrations of training details in the supplementary materials. To compare fairly with previous vision Mamba methods, our training settings strictly align with VMamba, whose training configurations were inherited from Swin Transformer. The training details for image classification are as follows: QuadMamba models are trained from scratch for 300 epochs, with a 20-epoch warm-up period, using a batch size of 1024. The training process utilizes the AdamW optimizer with betas set to (0.9, 0.999), a momentum of 0.9, an initial learning rate of $1\times 10^{-3}$, a weight decay of 0.05, and a cosine decay learning rate scheduler. Additional techniques such as label smoothing (0.1) and EMA (decay ratio of 0.9999) are also applied. No other training techniques are employed.
>
> For object detection and semantic segmentation, our method also strictly follows VMamba and Swin Transformer. The training details are as follows: Object detection and instance segmentation experiments are conducted on COCO 2017, which contains 118K training, 5K validation and 20K test-dev images. An ablation study is performed using the validation set, and a system-level comparison is reported on test-dev. For the ablation study, we consider four typical object detection frameworks: Cascade Mask R-CNN, ATSS, RepPoints v2, and Sparse RCNN in mmdetection. We utilize the same settings as in previous works: multi-scale training (resizing the input such that the shorter side is between 480 and 800 while the longer side is at most 1333), AdamW optimizer (initial learning rate of 0.0001, weight decay of 0.05, and batch size of 16), and $3 \times$ schedule (36 epochs).

---

> > ### Comment · Reviewer_ALic · 2024-08-13
> > **keep original score**
> >
> > All my concerns have been addressed. Thus, I keep my original score as strong accept.

---

### Official Review · Reviewer_Kbfx · 2024-07-11

**Soundness:** 3
**Presentation:** 3
**Contribution:** 3
**Rating:** 5
**Confidence:** 3

**Summary:**

Authors propose a technique to adapt Mamba to vision tasks. They propose a novel Quad-tree based approach instead of flattening tokens for images in a raster scan mechanism to avoid losing local dependencies. They evaluate on three vision tasks of object recognition, objet detection and instance & semantic segmentation.

**Strengths:**

- Interesting novelty and contribution designed specifically for Mamba type of models adapted to vision tasks.
- Practically beneficial algorithm on the efficiency side where the show lower FLOPs and parameters with better or on par performance to SOA

**Weaknesses:**

- It provides a tradeoff though and it's not clear which is more favourable when compared to VMamba with corresponding T/S/B in Table 1. Meaning you can see when comparing VMamba-T (88.2) with QuadMamba-T (78.2) it is much lower, it is only at the base variant they become on-par but with reduced FLOPS/parameters on QuadMamba side. The fact it doesn't seem that on the lighter weight variants the performance is on par in fact it is much lower with around 10%.

- What’s the reported inference time for Table 1 in addition to the FLOPs.

**Questions:**

- Table 2 not clear what is the result for LocalVMamba and VMamba with S for fair comparison to QuadMammba-S

---

> ### Author Rebuttal · Authors · 2024-08-06
>
> ### Q1.Clear and fair model comparisons
>
> Thanks for your valuable advice. We regret that our Li/T/S/B naming system may confuse readers compared with other methods. To clearly show our method's advantages, we rename the QuadMamba model variants (Lite -> B1, Tiny -> B2, Small -> B3, and Base -> B4). Our B1 and B2 variants are compared with efficient models, such as lite CNN models and efficient ViT models. Our B3 and B4 maintain similar model sizes as the tiny/small variants in transformer and vision Mamba backbone methods. Thus, our B1/B2 should be compared with those under 10M model parameters. Our B3 should be compared to tiny variants whose model parameters are around 30M, and our B4 to small variants around 50M. For instance, VMamba-T (9.1 GFLOPs) does not correspond to QuadMamba-S (5.5 GFLOPs) but QuadMamba-B (9.3 GFLOPs) instead. To make the comparison clearer, we highlight several comparisons under similar model sizes to show the advantages of the QuadMamba. Our model variants (Lite/Tiny) demonstrate clear superiority on extreme light model levels (under 10M). Moreover, our small and base models achieve comparable and better results compared to methods of similar model sizes.
>
> The detailed plots and comparisons are found in the attached PDF file. We highlight several comparisons under fair conditions to show the advantages of QuadMamba:
>
> | Model | #Params.(M) | Top1 Acc|
> |:------:|:------:|:------:|
> |Vim-Ti| 7| 76.1 |
> |LocalVim-Ti| 8| 76.2 |
> |PVT| 13.2| 75.1 |
> |QuadMamba-B1 (Lite)| 5.4| 74.2 |
> |QuadMamba-B2 (Tiny)| 10 | 78.2 |
> |Vim-S| 22| 80.5 |
> |LocalVim-S| 28| 81.2 |
> |QuadMamba-B3 (Small)| 31 | 82.4 |
>
>
> ### Q2.Reported inference time
>
> In Table 9 of the supplementary materials, we benchmark the throughput of the QuadMamba model variants on an A800 GPU platform. The results are also shown below. Compared to the vanilla VMamba model, our QuadMamba model has negligible inference latency. Moreover, the throughput of our lite and tiny models is much higher compared to the efficient CNN and Transformer models, which implies better inference efficiency.
>
> | Model|Renamed| #Params.(M) | Flops(G) | Throughput(img/s)|
> |:------:|:-------:|:------:|:------:|:------:|
> |QuadMamba-Li|B1| 5.4 | 0.8 | 1754 |
> |QuadMamba-T|B2| 10.3 | 2.1 | 1586 |
> |QuadMamba-S|B3| 31.2 | 5.5 | 1252 |
> |QuadMamba-B|B4| 50.6 | 9.3 | 582 |

---

### Official Review · Reviewer_HNK1 · 2024-07-14

**Soundness:** 3
**Presentation:** 3
**Contribution:** 3
**Rating:** 5
**Confidence:** 3

**Summary:**

The paper introduces QuadMamba, an enhancement of vision State Space Model (SSM) architectures. At its core is a learnable QuadVSS network block that processes the image input patch at two different resolutions. For every 2x2 coarse window with 4 image patches, the method adaptively learns to process one of the 4 patches at the finer 2x resolution using the differentiable Gumbel-softmax formulation. These block are hierarchically stacked similar to SwinTransformer, but using a window shifting scheme in two different directions which is a better fit for SSM.

The method is demonstrated to obtain good results relative to comparably sized transformer and CNN architectures on Imagenet classification, COCO object detection / instance segmentation, and ADE20k semantic segmentation.

**Strengths:**

- Intuitive idea to learn how to pick an area that requires higher resolution processing and pack it into SSM via Gumbel Softmax. Solid although not particularly novel stacking of QuadVSS blocks into a SwinTransformer-like network architecture.
- Competitive results relative to ViT and CNN, and seems to improve a little bit over other Mamba methods, although the gains there seem quite incremental.
- Ablation over multiple network parameter decisions.

**Weaknesses:**

# Significance
The contribution seems a bit incremental. The overall idea of image traversals into several windows to enforce better locality was already explored before in LocalMamba. The gains in the experimental section over some of the other Mamba network variants (EfficientVMamba, VMamba, Swin-S etc) are not too large. The idea that we do not have to process all areas in high resolution but only 1/4 of them does seem useful, although does not seem strictly limited to SSMs -- e.g. would ViT methods also benefit?

# Experimental results
- It seems that the natural baseline to this method is to compare a network made of VSS modules, as opposed to QuadVSS (the main novelty). Such a comparison was done in Fig 5, however it is not too clear how rigorous it is. It would be helpful to compare more directly VSS / QuadVSS equivalents with same FLOPS, for several different FLOPS thresholds. This does not seem to have been done - it would validate more strongly the fact that allocating parameters selectively to higher resolutions actually helps (as opposed to using the baseline VSS module pyramid).
- Unclear what the training overhead of a QuadVSS block is compared to a VSS block (in terms of memory, compute)?
- As opposed to just a table, it would be helpful to have plots with flops vs quality as axes, with a separate curve in that graph for each model family. Such a plot can make the comparison more obvious, given that different methods have somewhat different amounts of flops.
- In Sec 4.3, it's unclear why for the Tiny model, we stack more blocks in the second stage, while for Small and Base we stack more in the third stage. Any particular reason? Third stage is more standard VSS modules, as opposed to your QuadVSS innovation. Also later in Table 6 yet different stackings are best (2,4,6,2). It's hard to discern the logic in all these choices.

# Clarity and questions
Some details of the approach were not particularly clear to me.
- L175 Do you always pick the same fine-grained patch for all 2x2 windows, or do they vary by window? Not particularly clear from the exposition/notation.
- Also, it appears that 7 patches are picked for each region as per L185-187. Don't you want to have 8 patches (powers of 2 are usually more efficient hardware-wise?)
- The softmax in Eq 7, what labels do you train it on? Is it the final labels? Right now it appears that this part is trained first, and kept fixed.
- (minor) In Eq 6, it seems 'v_local' is aggregated across the whole image, shouldn't it be called 'global' instead?
- (minor) L186: where is 'local adjacency score' defined? This is the first mention of the term.
- (minor) Fig 5: S-QuadVSS is not defined, I assume it's the opposite direction shift, but helps to be explicit.

# Minor Nits and Typos
159: infOrmative
Table 1: VMamaba
606: Uperhead [62] (I believe it should be UperNet)

**Questions:**

- Is your quad idea limited to SSM or would it also work for Vit? Can one learn to downsample specific patches in a Vit?
- Do you have comparisons between networks made of QuadVSS/VSS or only VSS for several same FLOPS budgets? Do you have comparison of training overhead (compute/memory) for networks made of QuadVSS/VSS or only VSS baseline?
- L175 Do you always pick the same fine-grained patch for all 2x2 windows, or do they vary by window? Not particularly clear from the exposition/notation.
- Also, it appears that 7 patches are picked for each region as per L185-187. Don't you want to have 8 patches (powers of 2 are usually more efficient hardware-wise?)
- The softmax in Eq 7, what labels do you train it on? Is it the final labels? Right now it appears that this part is trained first, and kept fixed.

**Limitations:**

Limitations are adequately addressed.

---

> ### Author Rebuttal · Authors · 2024-08-05
>
> ### Q1.Significance
>
> **Q1A1 Comparing to LocalMamba:** How to effectively preserve 2D spatial dependencies is an important challenge in adapting sequence models into the vision domain. Though it has been partially explored by previous works such as PlainMamba [1] and LocalMamba [2], it is non-trivial that our proposed QuadVSS is a **data-adaptive** and **light-weight** design for the locality module.
>
> - Data-adaptive:  LocalMamba opts for a differential neural network search during training for the optimal locality strategy for each layer. Once trained,  the architecture of LocalMamba is fixed.
> For varied input images, LocalMamba used the same fixed locality strategy. In contrast, our QuadMamba has a learnable data-dependent locality strategy, which means it can dynamically and adaptively generate the optimal scanning sequences for different input data and various downstream tasks.
>
> - Light-weight: LocalMamba, which requires handcrafting the network search space and extra optimization losses, brings much more complexity during training.  Differently, our method brings minimal additional complexity in terms of training and optimization, as the quadtree-based scanning module is lightweight.
>
> **Q1A2 Gains and improvement to existing Mamba:** We believe this is a misunderstanding, and the gain of our method over existing ones is actually significant. We apologize for the possible slight confusion in Table 1's main results, which is due to misalignments between how our model variants and others (like VMamba) are named. For instance, VMamba-T (9.1 GFLOPs) does not correspond to QuadMamba-S (5.5 GFLOPs) but QuadMamba-B (9.3 GFLOPs) instead. To make the comparison clearer, we highlight several comparisons under similar model sizes to show the advantages of QuadMamba. Our model variants (Lite/Tiny) show clear superiority on extreme light model levels (under 10M). Moreover, our small and base models achieve comparable and better results compared to methods of similar model sizes. It is also worth noting that our method of bringing locality into Mamba is generalized, practical, and easy to implement. compared to other complex methods. More details are found in the attached PDF.
>
> **Q1A3 Applicable to ViT architecture:** Our quadtree-based window partition module and sequence construction strategy are specially designed for the recent vision Mamba models. The idea of our QuadMamaba originated from the coarse-to-fine feature representation philosophy, found in many prior arts such as InceptionNet [3], Multiscale Transformer [4], Focal Transformer [5], and Quad-attention Transformer [6]. However,  the casual sequence modeling scheme in the Mamba model is completely different from the non-casual attention scheme in vision transformers. The constructed sequence for Mamba has to casually scan each token from the input. The length of the token sequence in Mamba has to be kept the same for multi-layer feature processing.  Thus, it brings much difficulty in preserving spatial adjacency in Mamba. Differently, due to the flexibility of the attention scheme, the quantity of the output tokens can easily maintained if the query tokens remain unchanged. Thus, our method is highly customized for Mamba.
>
> ### Q2.Experimental results
>
> **Q2A1 More comparisons with the baseline:** To demonstrate the effectiveness of the proposed QuadVSS block, we will add additional ablation studies on different model sizes. The table below shows the improvement in tiny model levels/FLOPS thresholds.
>
> |Variant|Block|Params.(M)|Top-1 Acc.|
> |:------|:-------:|:------:|:------:|
> |Mamba-T|(2,6,2,2) |8.5 |76.9|
> |QuadMamba-T|(2,6,2,2)|10.2 |78.2|
>
>
> **Q2A2 Training overheads:** In the attached PDF files, we plot the GPU memory vs. Batch size and training curve. Our QuadMamba has an affordable GPU memory overhead compared to the baseline. We find that our method does not result in any difficulty in training convergence.
>
> **Q2A3 Plots with Flops vs. Performance:** We plot the detailed comparison of Flops and Performance in the attached PDF file. To clearly show the advantages of our method, we rename the QuadMamba model variants (Lite -> B1, Tiny -> B2, Small -> B3, and Base -> B4).
>
> **Q2A4 Block numbers in different stages:** According to our observation,  QuadVSS blocks work better in early model stages, where the feature resolution is higher. However, stacking more blocks in the second stage (high resolution) brings significant computation overload, especially for high-resolution images. To strike a balance of FLOPs between different stages, we allocate more QuadVSS blocks in the third stage (low resolution) instead of the second stage for larger models such as QuadMamba-Small and -Base (channels=144). For smaller models like QuadMamba-Lite and -Tiny, allocating more QuadVSS blocks to the second stage is more affordable because of fewer number of channels (channels=48), so it is unnecessary to defer the QuadVSS blocks to the third stage.
>
> ### Q3.Clarity and questions
>
> **Q3A1 Window hyper-parameter:** We always pick the 2x2 windows for two-level window partitions. Considering the 224x224 image resolution and four-time downsampling ratio, the two-level window partition strategy is enough for image classification. A more level window partition will be explored.
>
>
> **Q3A2 Patch hyper-parameter:** The image features, which are partitioned by learnable modules, are then reshaped to a 1D token sequence for the Mamba block. For the Mamba, the 1D token sequence is completely the same in terms of computation flows. Thus, the computing efficiency remains unchanged with different patch numbers.
>
> **Q3A3 Label of Eq.7:**  As the QuadMamba brings no extra training complexity, there is no training label for the softmax in Eq. 7. We design a differential sequence construction strategy and apply the Gumbel-Softmax trick, which can help optimize Eq.7.
>
>
> ### Q4.Minor nits and typos
>
>  We will revise the main text carefully.

---

> ### Comment · Reviewer_HNK1 · 2024-08-10
> **response**
>
> Thank you for the additional graphs and figures. While QuadMamba performs well compared to baselines in terms of #param, Fig 2 shows that performance is incremental wrt MACs (FLOPS). To me, actual model sizes are not as important, FLOPS (or model latency on same hw) are the primary criterion, and on this one the gains are quite small. So the graphs actually reinforce my initial statement - I do not think there is a naming misalignment - I was comparing similar flops to similar flops and there was not much notable improvement there.
>
> > Q2A1 The table below shows the improvement in tiny model levels/FLOPS thresholds.
>
> I do not see FLOPS listed in the table, just #params. I explicitly asked for quality comparisons for same FLOPS.
>
> > Q1A3
>
> Your answer seems to be "Mamba adds a bunch more complicated constraints to the process" but you did not actually answer my question of -- can I do sampling and apply the Gumbel-Softmax idea to ViTs?
>
> > As the QuadMamba brings no extra training complexity, there is no training label for the softmax in Eq. 7.
>
> Softmax usually suggests a classification objective with training data. It seems what actually is going on is Eq 9. But at the point where you introduce Eq 7, the context is missing and is quite confusing.
>
> I still lean positive but the explanations do not change my general takeaways, so I would keep my rating.

---

> ### Author Response · Authors · 2024-08-11
>
> Thanks for your responses and for maintaining positive ratings. We apologize for not addressing the concerns precisely due to the word limit. We hope the following explanation can partially address your concerns.
>
> * Our QuadMamba can achieve similar gains with a reasonable and simple learnable module compared to the complex methods (e.g., LocalViM).
>
> * We feel sorry that we forgot to provide the Flops in Q2A1. The Glops for Mamba-T is 1.7G, and the Flops for QuadMamba-T is 2.0G.
>
> Regarding Gumbel-Softmax:
>
> **1. Application to Vision Transformers (ViTs) and Other Architectures:**
>
> The Gumbel-Softmax technique can be integrated into Vision Transformers (ViTs)
> and various other architectures or tasks, such as detection and super-resolution.
> Unlike the Softmax, which outputs continuous values in the range (0,1), Gumbel-Softmax
>  provides discrete outputs in {0,1}. This capability is particularly beneficial for differentiable
> decision-making. Several ViT studies, like A-ViT[7] and SparseViT[8], utilize Gumbel-Softmax
> primarily for sparsifying feature computation. However, there's currently no exploration
> of using Gumbel-Softmax for patch partitioning in ViTs, which we believe is a promising
> area for future research.
>
> **2. Supervision and Learning:**
>
> Gumbel-Softmax here does not require explicit supervision.
> It receives gradients from the overall objective, such as the 1000-way cross-entropy loss
>  in ImageNet classification, and learns to predict coarse-to-fine partitions in an **end-to-end** manner.
> The supervision for the partitioning process is implicit.
>
>
> [7] A-ViT: Adaptive Tokens for Efficient Vision Transformer. Hongxu Yin, etc. In CVPR22.
>
> [8] SparseViT: Revisiting Activation Sparsity for Efficient High-Resolution Vision Transformer. Xuanyao Chen, etc. In CVPR23.

---

### Author Rebuttal · Authors · 2024-08-05

Dear Reviewers,

We thank the reviewers for the positive reviews of our work and constructive comments. Here is a list of  new figures and tables in the attached PDF file, and references referred to in other responses.

**Figures and tables** :

Figure 1. Plots of performance, mode size, and FLOPs in ImageNet classification.

Figure 2. Plots of performance, mode size, and FLOPs in COCO detection and ADE20K segmentation.

Figure 3. Training curve and GPU memory consumption of our method.

Table 1. Fair Comparisons in ImageNet classification.

**References** :

[1]  PlainMamba: Improving Non-Hierarchical Mamba in Visual Recognition. Chenhongyi Yang, etc. BMVC2024

[2] LocalMamba: Visual State Space Model with Windowed Selective Scan. Huang, etc. ArXiv2024.

[3] Going deeper with convolutions. Christian Szegedy, etc. Neurips2014

[4] Multiscale Vision Transformers. Haoqi Fan, etc. CVPR2021

[5] Focal Self-attention for Local-Global Interactions in Vision Transformers. Jianwei Yang, etc. Neurips2021

[6] QuadTree Attention for Vision Transformers. Shitao Tang, etc. ICLR2022

---

### Decision · Program_Chairs · 2024-09-25

**Decision:**

Accept (poster)

**Comment:**

The manuscript introduces a novel state space architecture for computer vision. The main novelty with respect to numerous previous attempts [26, 41, 49, 66, 80] (all only in Arxiv) corresponds to dynamic multi-resolution reasoning through learnable quadtree-based scanning. The reported performance is competitive with respect to prior state-space, convolutional and transformer models. Still, the overall per-MAC improvement is marginal and incremental at best.